# Micro- and Nanofibrillated Cellulose from Annual Plant-Sourced Fibers: Comparison between Enzymatic Hydrolysis and Mechanical Refining

**DOI:** 10.3390/nano12091612

**Published:** 2022-05-09

**Authors:** Roberto Aguado, Quim Tarrés, Maria Àngels Pèlach, Pere Mutjé, Elena de la Fuente, José L. Sanchez-Salvador, Carlos Negro, Marc Delgado-Aguilar

**Affiliations:** 1LEPAMAP-PRODIS Research Group, University of Girona, Carrer Maria Aurèlia Capmany 61, 17003 Girona, Spain; joaquimagusti.tarres@udg.edu (Q.T.); angels.pelach@udg.edu (M.À.P.); pere.mutje@udg.edu (P.M.); m.delgado@udg.edu (M.D.-A.); 2Department of Chemical Engineering and Materials, Universidad Complutense de Madrid, Avda. Complutense s/n, 28040 Madrid, Spain; helenafg@ucm.es (E.d.l.F.); josanc03@ucm.es (J.L.S.-S.); cnegro@ucm.es (C.N.)

**Keywords:** enzymatic hydrolysis, hemp, jute, mechanical pretreatments, nanocellulose, nanofibers, non-wood cellulose, sisal

## Abstract

The current trends in micro-/nanofibers offer a new and unmissable chance for the recovery of cellulose from non-woody crops. This work assesses a technically feasible approach for the production of micro- and nanofibrillated cellulose (MNFC) from jute, sisal and hemp, involving refining and enzymatic hydrolysis as pretreatments. Regarding the latter, only slight enhancements of nanofibrillation, transparency and specific surface area were recorded when increasing the dose of endoglucanases from 80 to 240 mg/kg. This supports the idea that highly ordered cellulose structures near the fiber wall are resistant to hydrolysis and hinder the diffusion of glucanases. Mechanical MNFC displayed the highest aspect ratio, up to 228 for hemp. Increasing the number of homogenization cycles increased the apparent viscosity in most cases, up to 0.14 Pa·s at 100 s^−1^ (1 wt.% consistency). A shear-thinning behavior, more marked for MNFC from jute and sisal, was evidenced in all cases. We conclude that, since both the raw material and the pretreatment play a major role, the unique characteristics of non-woody MNFC, either mechanical or enzymatically pretreated (low dose), make it worth considering for large-scale processes.

## 1. Introduction

There is little doubt that research on micro- and nanofibrillated cellulose (MNFC) has become a fertile and prolific field, with dedicated groups all around the world [1,2,3,4]. It could be said that, over the last decade, the field has been split into different branches. These include, to name a few, the production of gels and membranes, the reinforcement of packaging paper, the use of micro-/nanocellulose as rheology modifier, blending with different materials towards nanocomposites, and nanotechnology applications such as in optoelectronics and for nano-sized dielectric layers [1,5,6,7]. Another branch is based on exploring non-woody biomass for the production of MNFC [8,9,10]. Among the many alternative raw materials that have been considered for this purpose, we can mention as examples, kenaf [11], wheat straw [12], tobacco stalk, or bamboo [13]. Needless to say, this approach for production is compatible with virtually any of the aforementioned applications, even though they were generally proposed for nanocellulose from conventional raw materials (including packaging reinforcement and gels) [14,15]. However, measuring the morphological, rheological and surface charge properties is necessary to properly design experiments leading to innovative materials—a process that should start from the choice of the source of cellulose.

Generally speaking, the hypernym “nanocellulose” encompasses bacterial cellulose, cellulose nanocrystals or nanowhiskers, cellulose nanofibers, microfibrillated cellulose (even if their width is larger than 100 nm), and a mixture of the latter two [2,16]. It is widely considered that microfibrillated cellulose, by itself or along with some nanofibers, in what is called MNFC, is more industrially feasible than 100% nanofibrillated cellulose [17,18]. A nearly complete conversion towards nanofibers demands either an excessive energy input or an oxidizing pretreatment, whose scale-up is still challenging, despite some valuable recent contributions in that direction [19]. Mechanical pretreatments of cellulose and enzymatic hydrolysis are acknowledged as more suitable for large-scale manufacturing of MNCF [15,20]. This is not denying that true nanoscale cellulose is irreplaceable with microfibrillated cellulose when it comes to certain applications, such as optoelectronics or wound healing [21,22].

Previous works on the techno-economic assessment of MNFC production have highlighted the two methods that are, most probably, the most feasible as of today: an all-mechanical process and a mechanical fibrillation that is preceded by enzymatic hydrolysis [20,23,24]. In the latter case, the objective of the hydrolysis is to decrease the energy input, generally as electricity, that fibrillation requires, but at the expense of long reaction times. At a laboratory scale, both methods can be compared by keeping the fibrillation stage the same (e.g., high-pressure homogenization). This follows the pretreatment that is the object of study, be it pulp refining or beating to simulate the all-mechanical process [25], or depolymerization in the presence of endoglucanases [26].

Proving the suitability of certain non-woody sources for MNCF may give us another chance for the material valorization of lignocellulosic agricultural waste. A few decades ago, it seemed that papermaking was that chance, as some predictions estimated that the production of non-wood pulp in European countries would be over 5400 kt/year by 2010 [27]. Not only did reality fall below the worst-case scenario, as the actual production that year was 588 kt, but this number kept decreasing over time [28,29]. Furthermore, while second and third-generation biofuels offer a good opportunity [30], the European Council’s Waste Framework Directive prioritizes material recovery [31]. Therefore, manufacturers, users and lawmakers should welcome new contributions from the academic world in what pertains to non-woody crops as the source of cellulose.

In this work, we report the production and characterization of MNCF from jute, sisal and hemp after mechanical and enzymatic pretreatments. In a previous article of ours, nearly completely nanofibrillated cellulose was obtained from said raw materials, following oxidation of hydroxyl groups to carboxylates [32]. Despite the success of that process, we hereby suggest two approaches that, even though not for the same applications, are expected to be more feasible for production at an industrial scale. In this direction, enzymatically and mechanically pretreated chemical pulps were characterized in terms of morphology and composition. The MNCF samples generated therefrom had their aspect ratio, surface charge, and rheological behavior measured.

## 2. Materials and Methods

### 2.1. Materials

Bleached pulps from sisal, hemp and jute were provided by Celesa (Tortosa, Spain). Prior to mechanical or enzymatic pretreatments, 30 g of each pulp (on the basis of dry pulp weight) were dispersed in a pulp disintegrator at a consistency of 1.5%, for 20 min at 3000 rpm, in concordance with the ISO standard 5263 [33].

For the enzymatic hydrolysis, we used a 2% endo-β-1,4-glucanases commercial enzyme cocktail, Novozym 476 (Novozymes A/S, Bagsværd, Denmark), with an activity factor of 4500 CNF-CA/g cellulose. Polydiallyldimethylammonium chloride (polyDADMAC) was supplied by L.C. Paper (Besalú, Spain), while a sodium polyethylene sulfonate (PES-Na) aqueous solution was provided by BTG Instruments (Säffle, Sweden). The reagents required for the chemical characterization of pulps, such as those for any other test mentioned from now on, were purchased from Sigma-Aldrich (Barcelona, Spain).

### 2.2. Pretreatments

For the glucanase-mediated hydrolysis, the aforementioned enzyme cocktail was added to the aqueous suspension of the pulp, under gentle agitation, at dosage levels of 80 and 240 mg/kg. The suspension was stirred for 2 h at 50 °C. Afterwards, enzymes were deactivated by heating the suspension at 80 °C for 15 min. The treated pulp was rinsed with distilled water [34].

For the mechanical pretreatment, the pulp suspension was filtered, adjusting its consistency to 10 wt.%, and then refined to 20,000 revolutions in a PFI mill NPFI 02 from Metrotec S.A. (Lezo, Spain), which complies with ISO 5264–2:2002 [25,33].

### 2.3. Characterization of Pretreated Pulps

The original pulps and samples after each of the pretreatments were subjected to the common TAPPI test methods for raw materials and/or pulps [35]. The samples for analysis were prepared according to T 264 cm-07. Solid-liquid extractions followed T 204 cm-17 for ethanol-benzene extractives, while the ash content was determined by means of a muffle furnace in accordance with T 211 sp-11. The test for the determination of acid-insoluble (Klason) lignin was carried out with H_2_SO_4_ 24N (T 222 om-15), whereas acid-soluble lignin was computed from the UV absorbance of the filtrate from the former test (TAPPI UM 250).

The morphological analysis of fibers was carried out by means of a MorFi Compact Analyzer (TechPAP, Gières, France), whose image analysis system involves the software MorFi v9.2 [36]. Besides, the content of fines in weight was measured by filtering the pulp suspension at a consistency of 1% through a 200-mesh screen, corresponding to a sieve opening of 75 μm.

### 2.4. Fibrillation

Each of the nine pretreated pulps, resulting from the three kinds of pretreatments on the three original pulps, was diluted to 1 wt.% consistency. Then, they were subjected between 3 and 9 times to high-pressure homogenization (HPH) in a laboratory scale homogenizer NS1001

PANDA 2K-GEA (GEA Niro Soavi, Parma, Italy). The common-to-all treatment consisted of 3 passes at 300 bar, based on previous experience [10]. Up to four more sets of MNFC were produced by additional HPH cycles at 600 or 900 bar. Figure 1, besides schematizing the general experimental layout, shows the sample code for each of the fibrillation sequences.

### 2.5. Characterization of Micro- and Nanofibrillated Cellulose

The yield of nanofibrillation was estimated by diluting a CNF suspension to 0.2 wt.% and centrifugation for 20 min at 4500 rpm (1254× *g*) in a Sigma Laborzentrifugen device, model 6K15 (Osterode, Germany). The sediment, essentially consisting of the microfibrillated (not nano-) part, was oven-dried until a constant weight and weighed. Thus, the yield is expressed as the mass ratio of NFC, found in the supernatant, to MNFC, always on a dry weight basis. Submitting every well-mixed MNFC suspension to spectrophotometry in the visible range, performing the reading before any settling can be appreciated, provides an idea of the size of dispersed nanofibers, considering particle-induced light scattering phenomena [37]. Hence, a spectrophotometer from Shimadzu, model UV-160A, was set to display transmittance and the value at a wavelength of 600 nm was recorded for all samples.

The average aspect ratio of MNFC (Equation (1)) was estimated from the gel point (Equation (2)), as originally suggested by Prof. Batchelor’s group [38] and simplified in recent works [39].
(1)$$\mathrm{Aspect}\;\mathrm{ratio}=6.0\;{\left(\frac{\mathrm{Gel}\;\mathrm{point}}{1000}\right)}^{–0.5}$$
(2)Gel point=dϕd(HsH0)≈ϕiHs,iH0

Briefly, the differential equation implied by the gel point method is approximated to the quotient between the initial concentration (*ϕ_i_*) and the initial variation in the sediment height ratio (*H_s,i_/H_0_*). For this test, each MNFC suspension was diluted to a certain initial concentration (*ϕ_i_* ~ 0.1–0.3 wt.%) and shaken for 10 min, in such way that *H_s,i_/H_0_* lies between 0.04 and 0.12. We added 200 μL of crystal violet (0.1 wt.%) to ease visualization of the sediment. Then, 250 mL of this mixture was left undisturbed in a graduated probe, until sedimentation was finished.

The rheological behavior of MNFC suspensions was assessed with a PCE-RVI 2 V1L rotational viscometer (PCE Instruments, Hamburg, Germany), rotating at 0.3–200 rpm to attain a shear rate range of 0.2–140 s^−1^. Apparent viscosity results were fitted to the shear rate applied in each case, according to the Ostwald-de Waele equation:(3)$$\eta =k\;\gamma^{n-1}$$
where *k* is the consistency factor and *n* is the flow behavior index.

The remaining carboxyl group content (mostly from hemicellulose) was estimated by conductimetric titration [25]. A potentiometric titration by means of a particle charge detector from BTG Instruments (Säffle, Sweden), Mütek PCD 04, allowed us to measure the cationic demand (*CD*). For that, each MNFC sample was soaked in excess polyDADMAC and a back titration with a polyelectrolyte of the opposite sign, PES-Na, was carried out [32].

Datasets were compared in terms of paired t-tests to discern whether or not a difference is significant, opting for a significance level of α = 0.05 [40]. The null hypothesis is defined as follows: the average of the differences between two sets of values is zero. A significant difference is considered to be attained if the calculated *p*-value is less than 0.05.

## 3. Results and Discussion

### 3.1. Pretreated Pulps from Jute, Sisal, and Hemp

As expected for bleached chemical pulps, the main constituent of the starting materials is cellulose, following solubilization (when not direct dissolution) of most hemicelluloses, lignin, lipophilic extractives, and inorganic compounds. The composition of each of the pulps is depicted in Figure 2. None of the pretreatments exerted significant effects on the ratio of cellulose to any of the other components (Table A1). It is inferred, therefore, that generation of soluble reducing sugars (cellobiose, cellotriose) by mechanical and enzymatic pretreatments can be safely neglected. Indeed, mechanical stress is generally assumed to break down polymers at the middle of the chain, not at the ends [41], and so do ‘endo’-acting enzymes, such as the ones included in the cocktail used in this work [20].

Nonetheless, a clear effect of both mechanical stress and enzymatic hydrolysis of cellulose is fiber shortening. The impact of said pretreatments on the average length is shown in Figure 3 (weighted in length) and in Table 1 (arithmetic mean length). The length-weighted average is more insensitive to the values at the extremes and is often found to be more easily correlated to mechanical properties [36], while the abundance in very short fibers results in the arithmetic mean length being considerably lower [42].

Shear forces may be a direct cause for fiber disruption, while there are two major hypotheses, not incompatible with each other, for the mechanism of glucanases. In some works, enzymatic damage to fibers is alleged to be mostly a result of “peeling”, i.e., hydrolysis takes place primarily at the surface [43]. It is also frequently stated that glucanases target selectively the amorphous domains of cellulose [44]. According to this, within a hemicellulose-containing fiber, the β-1,4 bonds of glucomannans would be a favored target for enzymatic hydrolysis, and thus their cleavage may favor the separation of fibrils. That said, it is reasonable to imply that such a phenomenon happens preferably at the surface of the fibers (external fibrillation). In any case, regarding the effects on fiber length, there was no significant difference between the different pretreatments (0.05 < *p* < 0.31), but all of them significantly shortened fibers.

Moreover, the percentage of fines increased, as they were mostly generated from fibers. At the same time, despite the peeling and disruption of the fiber wall, there were no appreciable effects on fiber width, indicating that erosion was compensated by swelling. This swelling effect, concomitant with internal fibrillation, is also the reason why the coarseness decreases (Table 1), as the density of water is lower than that of dry pulp (approximately 1.5 g·cm^−3^). Internal fibrillation was seemingly more accentuated in the case of sisal pulp refining.

### 3.2. Mechanical Micro- and Nanofibers

Some key properties of mechanical MNFC, i.e., the material that results from fibrillating severely refined chemical pulps, can be found in Table 2 and in Table A2 (Appendix A). Since the carboxyl content and the cationic demand are, by themselves, of little interest for mostly electrically neutral fibrils, they were used to estimate the specific surface area (*SSA*) as developed in a previous work [32]:(4)$$SSA=\left(CD-CC\right)\times 0.155{\;m}^{2}/\mu \mathrm{eq}$$
where both the cationic demand (*CD*) and the carboxyl content (*CC*) are given in µeq/g.

*CD* is due to the remaining carboxyl groups from hemicellulose’s glucuronic acid units (also quantified in Table 2) and to polarized hydroxyl groups [32]. By fibrillation and subsequent increase in the specific surface area, more functional groups of all kinds become exposed to the cationic polyelectrolyte (polyDADMAC) during the first step of the measurement of the cationic demand through potentiometric back titration. The surface of microfibrils is readily available for this ionic exchange, but the highly organized macromolecular assembly of cellulose and hemicellulose chains hinders the diffusion of polymers through the wall, especially if the molecular weight of said polymers is high [45]. In this case, the average molecular weight of polyDADMAC (107 kg/mol) fulfills this condition. As microfibers are peeled off, eroded and disrupted towards nanofibers, these limitations to mass transfer decrease. Consistently, albeit with two exceptions, more passes through the homogenizer are translated into higher aspect ratio and higher *SSA*.

By the aforementioned mechanisms, as we repeatedly passed the suspension of microfibers through the high-pressure pump, nanofibers were incrementally generated from microfibers. The trend is evidenced in Figure 4. In each case, more transparency (more transmittance) means that the pulp was fibrillated to a larger extent. Not surprisingly, transmittance is tightly correlated to the yield of nanofibrillation (Figure 4). After three HPH cycles at 300 bar, MNFC suspensions were still almost opaque. Nine passes through the pump and valve of the homogenizer resulted in notoriously higher transparency, but still, less than 20% of the light (in terms of intensity) passed through the cuvette. In contrast, nanofibrillated cellulose from these same raw materials, obtained after TEMPO-mediated oxidation and then homogenization, may reach transmittance values above 95% [10]. In that case, the repulsion introduced by the chemical modification eased the nearly complete disruption of the fiber. However, with no other modification rather than limited depolymerization, the disintegration of fibers to the nanoscale was not achieved, regardless of the raw material. Likewise, the effect of the HPH steps was statistically independent of the raw material, finding no significant difference among jute, sisal, and hemp (0.26 < *p* < 0.83).

### 3.3. Enzymatic Micro- and Nanofibers

Not unlike mechanical MNFC, micro- and nanofibers that were produced after a pretreatment consisting of enzymatic hydrolysis show increasing *CD* (Table A3), and thus increasing *SSA*, with an increasing number of HPH passes. This is shown in Table 3, which also indicates that the evolution of the aspect ratio was not as consistent as in the case of mechanically pretreated MNFC. There were no significant differences in what pertains to the carboxyl content as compared to mechanical MNFC (Table A3).

Overall, the aspect ratio attained by enzymatic MNFC is lower than that of mechanical MNFC. This replicates, although more markedly, the results for micro- and nanofibers from aspen wood, whose aspect ratio was found in a previous work to decrease in the order: mechanical pretreatment > enzymatic hydrolysis > TEMPO-mediated oxidation [46]. The easiest explanation lies in how fibers are before starting the homogenization in each case. Refining causes both external and internal fibrillation, easing the separation of microfibrils that are further peeled off by HPH. Enzymatic hydrolysis also peels the fiber wall preferably, but the cleavage of chains from cellulose and the β-glucans in the remaining hemicellulose unavoidably causes significant shortening of microfibrils. Finally, oxidizing cellulose largely promotes a nearly complete disruption of microfibrils towards nanofibers (by subsequent HPH). In addition, as oxidized nanofibers are rich in carboxylate groups (highly hydrophilic), they easily undergo swelling in water. Indeed, the aspect ratio of TEMPO-oxidized nanofibers from sisal, jute and hemp was found to be lower than that of the MNFCs studied in this work, corroborating the aforementioned sequence of pretreatments [10].

Regarding the yield of nanofibrillation (*p* = 2 × 10^−7^) or the transmittance (*p* = 10^−4^), the enzyme dosage imparted significant differences. However, increasing the dosage of endoglucanases from 80 to 240 mg/kg did not exert a notorious influence on *SSA*, and no significant influence was observed in the aspect ratio (*p* = 0.23). These properties follow the same trend as the yield of fibrillation towards nanofibers and the tightly related transmittance of enzymatic MNFC, which are presented in Figure 5. The fact that a triplication of the enzyme dosage achieves only a timid enhancement supports at least one of the two complementary hypotheses on the mechanism of glucanase-mediated hydrolysis—that domains with a high degree of supramolecular order (crystallinity) are particularly resistant to enzymatic cleavage. Thus, the process seems to be controlled by limitations in the diffusion of the enzyme through highly ordered (by hydrogen bonding) regions.

### 3.4. Effect of Fibrillation Energy on Rheological Behavior

As the suspension of microfibrils gradually shifted towards a dispersion of nanofibers by increasing the number of HPH cycles, the apparent viscosity at a given shear rate (100 s^−1^) increased in most cases, as plotted in Figure 5 as a function of the energy input. The correspondence of the electric power reading (using Socomec’s Diris A-20, Barcelona, Spain) to each sequence is displayed in Table 4 and shown in more detail, step-by-step, in the Appendix A (Table A4). Regarding the differences between raw materials, they are significant, and it could be said that MNFC from jute was found to be consistently more viscous than that from sisal (*p* = 8 × 10^−5^) and that from hemp (*p* = 0.015).

During homogenization, the mean particle size of the dispersed phase, consisting of micro- and nanofibrils, decreases, while their number in a given element of volume increases. Micro- and nanofibers immobilize more water molecules (continuous phase) since, as more area is exposed, particle–solvent and particle–particle interactions are enhanced. Thus, the apparent viscosity tends to increase, but this is a simplified approach [47]. This conclusion could not have been safely drawn *a priori*, since the viscosity of nanocolloids is actually reported to increase with increasing particle size [48]. While the suspensions assessed in this work could not be classified as nanocolloids, as microscale cellulose outweighs nanoscale cellulose, heterogeneity hampers any attempt of prediction. In fact, Figure 6 shows some exceptions to the increasing trend, particularly for MNFC from sisal pulp (Figure 6b). Furthermore, the low increment shown by enzymatic MNFC (low dosage) from hemp pulp (Figure 6c) cannot be deemed significant. All things considered, the apparent viscosity of heterogeneous MNFC suspensions is led by different mechanisms and experimentation is required in each case.

Besides the energy input, Table 4 also shows the consistency factor and the flow behavior index of the Ostwald–de Waele equation (Equation (3)) for MCNF from jute, sisal and hemp, after undergoing refining. For the fitting parameters of enzymatic MCNF, the reader is referred to the Appendix A (Table A1 and Table A2). While the consistency factor consistently increases with progressive fibrillation, the flow behavior index decreases until approaching a constant value. In other words, the shear-thinning or pseudoplasticity behavior becomes more marked by increasing the proportion of nanofibers, particularly in the case of sisal MNFC, but little or no difference is observed from seven to nine HPH cycles. Cellulose suspensions, even when not fibrillated, are known to display a shear-thinning or pseudoplastic (*n* < 1) nature [49], and fibrillation enhances this effect by increasing the surface area that can interact with water. For micro- and nanocellulose, a shear-thinning behavior has been consistently reported [10,50], but, as shown in this work, the extent or degree of this behavior is highly dependent on the starting materials and on the treatments they are subjected to.

## 4. Conclusions

It has been evidenced that the composition and fiber morphology of chemical pulps from jute, sisal and hemp make these pulps solid candidates for the production of cellulose micro-/nanofibers. Both PFI refining and enzymatic hydrolysis were suitable pretreatments for the subsequent high-pressure homogenization, although in the latter case, a low dosage of enzyme (80 mg/kg) is recommended for the sake of feasibility. Micro-/nanofibers with a high aspect ratio were obtained, especially in the case of the mechanical pretreatment and when hemp pulp was the starting material. Mechanical MNFC from jute also resulted in more viscous dispersions (up to 140 mPa·s at 1% consistency and 100 s^−1^), observing increasing apparent viscosity with an increasing number of homogenization cycles. Moreover, the enzymatically pretreated jute pulp was the easiest to fibrillate towards nanofibers, thus requiring a lower energy input for a certain degree of nanofibrillation.

This work cannot conclude, nonetheless, whether one source should be recommended over another in general terms, either among the three raw materials involved or in comparison to wood pulp. Instead, we show that key characteristics of MNFC, such as the aspect ratio, the surface area and the rheological behavior, depend not only on the production process, but also (and strongly) on the choice of starting material. Therefore, depending on the specific application of micro-/nanofibers, decision making in the near future should consider, besides the conventional wood pulps, pulps from non-wood crops for attaining a broader spectrum of possibilities.

## Figures and Tables

**Figure 1 nanomaterials-12-01612-f001:**
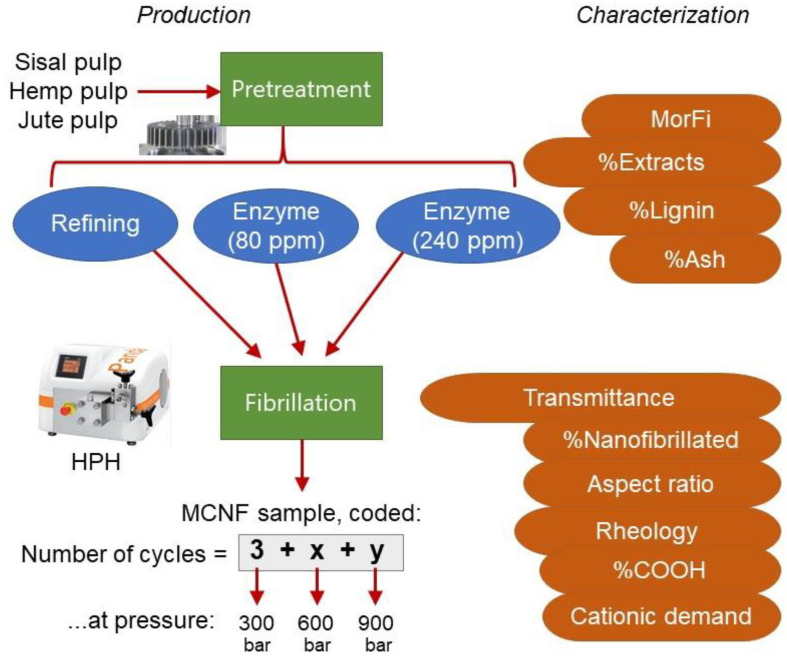
General experimental layout.

**Figure 2 nanomaterials-12-01612-f002:**
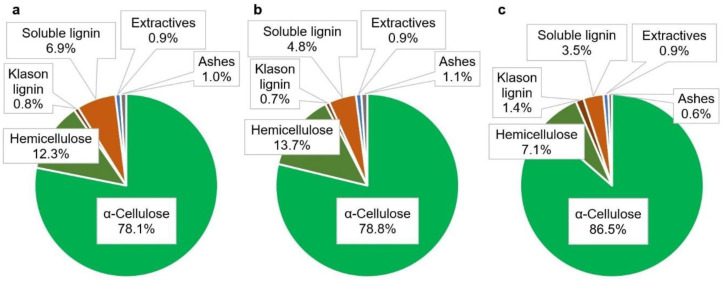
Chemical composition of the pulps from jute (**a**), sisal (**b**), and hemp (**c**).

**Figure 3 nanomaterials-12-01612-f003:**
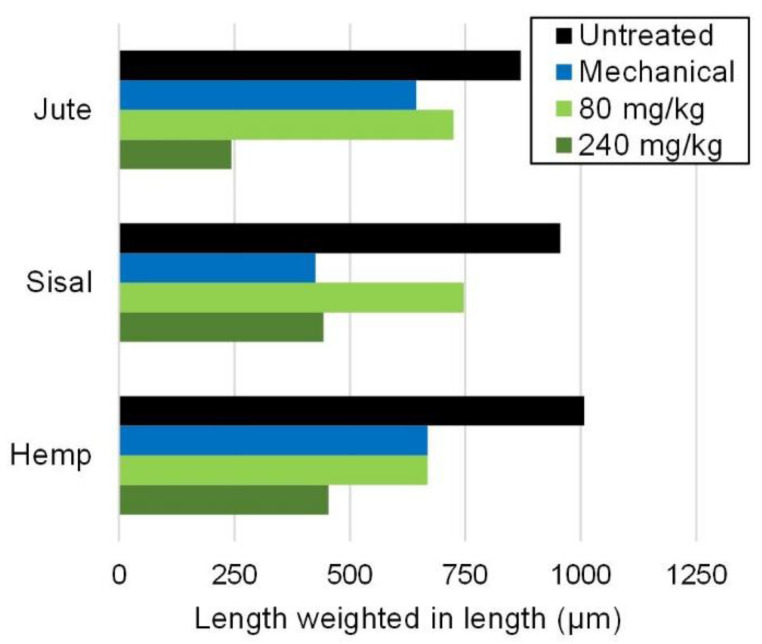
Effect of mechanical and enzymatic pretreatments on fiber length, expressed as the length-weighted mean.

**Figure 4 nanomaterials-12-01612-f004:**
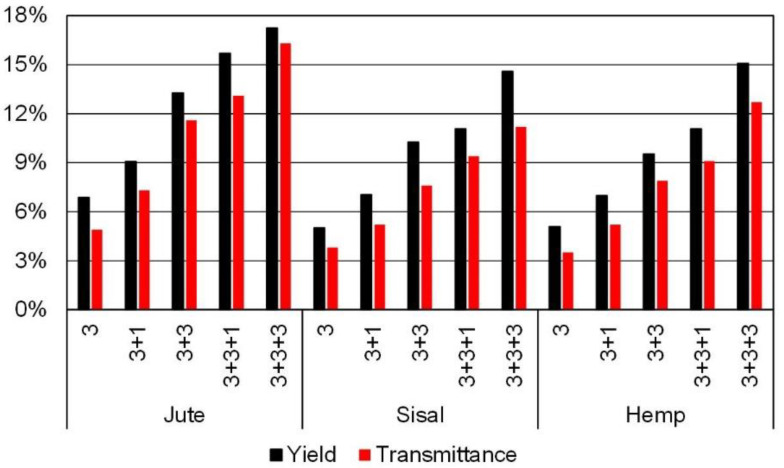
Yield of nanofibrillation and transmittance at 600 nm for jute, sisal and hemp MNCF after extensive PFI refining and different numbers of HPH cycles.

**Figure 5 nanomaterials-12-01612-f005:**
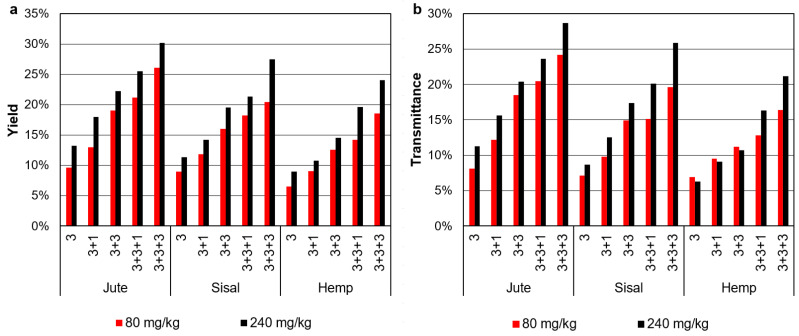
Yield of nanofibrillation (**a**) and transmittance at 600 nm (**b**) for jute, sisal and hemp MNCF whose pretreatment consistent of an endoglucanase-mediated hydrolysis at different dosages. The horizontal axis indicates the HPH sequence: cycles at 300 bar, at 600 bar and at 900 bar.

**Figure 6 nanomaterials-12-01612-f006:**
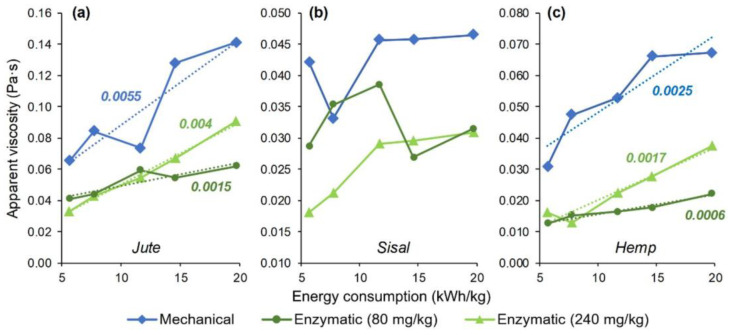
Evolution of the apparent viscosity at 100 s^−1^ of suspensions of MNFC from jute (**a**), sisal (**b**) and hemp (**c**). Numbers in italics represent the slope of a trend line.

**Table 1 nanomaterials-12-01612-t001:** Effects of mechanical and enzymatic pretreatments on key parameters of fiber morphology.

Pulp	Pretreatment	Arithmetic Length (µm)	Diameter (µm)	Coarseness (mg/m)	Fines (%)
**Jute**	Untreated	566	24.6	0.249	26.3
	Mechanical	440	23.2	0.141	50.3
	Enzymatic, 80 mg/kg	492	24.1	0.177	29.5
	Enzymatic, 240 mg/kg	175	21.8	0.160	37.3
**Sisal**	Untreated	449	21.6	0.353	21.6
	Mechanical	246	21.3	0.190	45.3
	Enzymatic, 80 mg/kg	287	21.1	0.265	25.3
	Enzymatic, 240 mg/kg	246	21.1	0.314	39.3
**Hemp**	Untreated	551	25.4	0.370	27.8
	Mechanical	279	21.8	0.349	40.2
	Enzymatic, 80 mg/kg	309	22.8	0.355	32.8
	Enzymatic, 240 mg/kg	248	24.1	0.305	44.1

**Table 2 nanomaterials-12-01612-t002:** Influence of the number of HPH cycles on the specific surface area and the aspect ratio of MNFCs that have undergone a mechanical pretreatment.

	Cycles	*SSA* (m^2^/g)	Aspect Ratio
**Jute**	3	16.7	144
	3 + 1	18.6	147
	3 + 3	20.9	156
	3 + 3 + 1	23.4	163
	3 + 3 + 3	25.1	164
**Sisal**	3	13.8	117
	3 + 1	16.0	123
	3 + 3	18.8	139
	3 + 3 + 1	21.5	139
	3 + 3 + 3	24.3	152
**Hemp**	3	12.4	188
	3 + 1	14.7	211
	3 + 3	17.7	193
	3 + 3 + 1	20.6	228
	3 + 3 + 3	23.4	217

**Table 3 nanomaterials-12-01612-t003:** Effect of the number of HPH cycles on the specific surface area and the aspect ratio of MNFCs that underwent enzymatic hydrolysis.

Dose of Enzyme (mg/kg)	Cycles	Jute		Sisal		Hemp	
		*SSA* (m^2^/g)	Aspect Ratio	*SSA* (m^2^/g)	Aspect Ratio	*SSA* (m^2^/g)	Aspect Ratio
**80**	3	21.5	80	19.5	64	15.0	83
	3 + 1	23.9	92	21.2	64	17.5	86
	3 + 3	25.0	90	23.6	67	20.2	87
	3 + 3 + 1	26.0	89	25.6	74	22.8	97
	3 + 3 + 3	26.8	97	26.7	78	24.2	103
**240**	3	24.3	89	20.8	64	16.7	81
	3 + 1	25.4	86	23.4	64	20.9	84
	3 + 3	27.3	79	25.0	66	22.6	89
	3 + 3 + 1	27.7	82	26.4	61	23.9	99
	3 + 3 + 3	28.8	86	28.5	63	25.9	103

**Table 4 nanomaterials-12-01612-t004:** Energy consumption and Ostwald–de Waele fitting parameters, both the consistency factor (*k*) and the flow behavior index (*n*), for mechanical MCNF.

Cycles	Energy Consumption (kWh/kg)	Jute		Sisal		Hemp	
		*k* (Pa·s^−*n*^)	*n*	*k* (Pa·s^−*n*^)	*n*	*k* (Pa·s^−*n*^)	*n*
**3**	155	1.44	0.17	1.63	0.21	1.06	0.40
**3 + 1**	167	2.36	0.15	1.98	0.11	1.42	0.39
**3 + 3**	182	2.66	0.15	3.28	0.07	1.64	0.33
**3 + 3 + 1**	198	4.02	0.11	3.31	0.07	2.80	0.33
**3 + 3 + 3**	209	4.22	0.10	3.49	0.06	3.06	0.33

## Data Availability

Available at the repository of the University of Girona, https://dugi-doc.udg.edu/ (accessed on 5 May 2022).

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
