# Peer review of "Micro- and Nanofibrillated Cellulose from Annual Plant-Sourced Fibers: Comparison between Enzymatic Hydrolysis and Mechanical Refining"

_nanomaterials, 2022, doi:10.3390/nano12091612_

Round 1

Reviewer 1 Report

This work assesses a technically feasible approach for the production of micro- and nanofibrillated cellulose (MNFC) from jute, sisal, and hemp, involving refining and enzymatic hydrolysis as pretreatments. Regarding the latter, only slight enhancements of nanofibrillation, transparency and specific surface area were recorded when increasing the dose of endoglucanases from 80 to 240 mg/kg. This supports the idea that highly ordered cellulose structures near the fiber wall are resistant to hydrolysis and hinder the diffusion of glucanases. Mechanical MNFC displayed the highest aspect ratio, up to 228 for hemp. Increasing the number of homogenization cycles increased the apparent viscosity in most cases, up to 0.14 Pa·s at 100 s-1 (1 wt.% consistency). I considered it can be published in our journal after a minor revision.

  1. The figures and tables in the text and the corresponding figures and tables should be centered. The font in Figure 4 is not consistent with that in other pictures, so the author is suggested modifying it and unifying the font.
  2. Fig. 5 is too vague. I suggest the author redraw it to improve the clarity. In the introduction, the author mentions “It could be said that, over the last decade, the field has been split into different branches. and does not specifically list which branches this research field is divided into.” It is suggested that the author list several branches and cite relevant literature to enrich this study.
  3. In the introduction, the author mentions “but measuring the morphological, rheological and surface charge properties is necessary to further design experiments leading to innovative materials.” Please illustrate the necessity of these properties for material preparation. “Internal fibrillation was seemingly more accentuated in the case of sisal pulp refining.” Why does the author say so in this paper? It is suggested to list several previous studies to further enrich this study.
  4. It is suggested that all the pictures in the article are redrawn and color-matched. What is the innovation point of this paper? I hope the author will introduce the innovation point of this paper and give appropriate examples to prove it. And the author should compare the previous work in this area to highlight the rationality and innovation of this paper. It is suggested that the author reorganize the abstract part and the conclusion part to express the gist and innovation of this paper.
  5. The author must carefully check the statements in the manuscript and correct the mistakes pointed out in it. add some references such as “Composites Science and Technology, 221 (2022) 109178., Chemical Engineering Journal, 2022, 431(1): 133919."

Author Response

This work assesses a technically feasible approach for the production of micro- and nanofibrillated cellulose (MNFC) from jute, sisal, and hemp, involving refining and enzymatic hydrolysis as pretreatments. Regarding the latter, only slight enhancements of nanofibrillation, transparency and specific surface area were recorded when increasing the dose of endoglucanases from 80 to 240 mg/kg. This supports the idea that highly ordered cellulose structures near the fiber wall are resistant to hydrolysis and hinder the diffusion of glucanases. Mechanical MNFC displayed the highest aspect ratio, up to 228 for hemp. Increasing the number of homogenization cycles increased the apparent viscosity in most cases, up to 0.14 Pa·s at 100 s-1 (1 wt.% consistency). I considered it can be published in our journal after a minor revision.

First of all, we are grateful to the reviewer for considering our manuscript and fot the helpful comments.

The figures and tables in the text and the corresponding figures and tables should be centered. The font in Figure 4 is not consistent with that in other pictures, so the author is suggested modifying it and unifying the font.

Centering: fixed. Fig. 4: hopefully fixed. It was Arial too, but the different way in which the figure was inserted ("Insert" vs. "Paste as enhanced metafile") caused some distortion.

Fig. 5 is too vague. I suggest the author redraw it to improve the clarity.

It has been redrawn, and split into two ("a" and "b") for the revised version.

In the introduction, the author mentions “It could be said that, over the last decade, the field has been split into different branches". and does not specifically list which branches this research field is divided into. It is suggested that the author list several branches and cite relevant literature to enrich this study.

Addition to the revised version of the manuscript: "These include, to name a few, the production of gels and membranes, the reinforcement of packaging paper, the use of micro-/nanocellulose as rheology modifier, blending with different materials towards nanocomposites, and nanotechnology applications such as in optoelectronics and for nano-sized dielectric layers [1,5–7]."

It is suggested that all the pictures in the article are redrawn and color-matched.

Black & Red is now the typical pattern of the manuscript's charts.

What is the innovation point of this paper? I hope the author will introduce the innovation point of this paper and give appropriate examples to prove it. And the author should compare the previous work in this area to highlight the rationality and innovation of this paper. It is suggested that the author reorganize the abstract part and the conclusion part to express the gist and innovation of this paper.

At this point, it is difficult and it would be surprising to find a suitable lignocellulosic raw material that has not been used for nanocellulose production yet. We are aware that micro- and nanofibers have already been obtained from hemp, jute, and sisal, but it is the sum of different approaches that turn this manuscript into a useful piece within the complex puzzle of non-wood nanocellulose. The particular focus in dimensions+transparency+rheology (a trademark of our group), the comparison between enzymatical and mechanical pretreatment, and the comparison between angiosperm plants from different orders towards the same product, will be useful for many researchers and/or institutions. Hopefully, the revised version threads this concept with the previous efforts in a better way.

The author must carefully check the statements in the manuscript and correct the mistakes pointed out in it. add some references such as “Composites Science and Technology, 221 (2022) 109178., Chemical Engineering Journal, 2022, 431(1): 133919."

Corrections are marked in red in the revised version. Yang et al.'s work has been cited.

Reviewer 2 Report

Study on preparing micro- and nanofibrillated cellulose is useful for utilizing the biomaterials. Two methos was compared in this article.However, the writting and organization of the article need to be greately improved. Some questions are listed as follows.  

1 Mechanical and enzymatic pretreatment are the commonly used methods, the authors need to introduce these studies in Introduction.

2 What’s the purpose application of micnanofibral cellulose?

3 Figure 2, the content of cellulose from hemp is missing.

4 Line 220, “Nine passes through the pump and valve of the homogenizer resulted in notoriously higher transparency, but still, less than 20% of the light (in terms of intensity) passed through the cuvette. In contrast, nanofibrillated cellulose from these same raw materials, obtained after TEMPO-mediated oxidation and then homogenization, may reach transmittance values above 95% [7].” Please explain the reason.

5 Line 252-271, The results got from Figure 5 is not accurate. Significance analysis of difference should be adopted.  

6 Line 281-284, I think Figure 5 should be Figure 6.

7 Line 290, I did not find Table 5

8 All data in the article should be analyzed based on the significance of diffrence to ensure the correct conclusion.

9 Writing need to improve.

Author Response

Study on preparing micro- and nanofibrillated cellulose is useful for utilizing the biomaterials. Two methos was compared in this article.However, the writting and organization of the article need to be greately improved. Some questions are listed as follows.  

1 Mechanical and enzymatic pretreatment are the commonly used methods, the authors need to introduce these studies in Introduction.

Line 57 in the revised version: "Previous works on the techno-economic assessment of MNFC production have highlighted the two methods that are, most probably, the most feasible as of today: an all-mechanical process and a mechanical fibrillation that is preceded by enzymatic hydrolysis [19,22,23]. In the latter case, the objective of the hydrolysis is to decrease the energy input, generally as electricity, that fibrillation requires, but at the expense of long reaction times. At a laboratory scale, both methods can be compared by keeping the fibrillation stage the same (e.g., high-pressure homogenization). This follows the pretreatment that is object of study, be it pulp refining or beating to simulate the all-mechanical process [24], or depolymerization in the presence of endoglucanases [25]."

2 What’s the purpose application of micnanofibral cellulose?

A list of applications is now provided in the introduction (line 32): "...the production of gels and membranes, the reinforcement of packaging paper, the use of micro-/nanocellulose as rheology modifier, blending with different materials towards nanocomposites, and nanotechnology applications such as in optoelectronics and for nano-sized dielectric layers [..."

3 Figure 2, the content of cellulose from hemp is missing.

This has been fixed now (86.5%).

4 Line 220, “Nine passes through the pump and valve of the homogenizer resulted in notoriously higher transparency, but still, less than 20% of the light (in terms of intensity) passed through the cuvette. In contrast, nanofibrillated cellulose from these same raw materials, obtained after TEMPO-mediated oxidation and then homogenization, may reach transmittance values above 95% [7].” Please explain the reason.

An explanation is given in the revised version, right after the quote highlighted by the reviewer: 

"In that case, the repulsion introduced by the chemical modification eased the nearly complete disruption of the fiber. However, with no other modification rather than limited depolymerization, the disintegration of fibers to the nanoscale was not achieved, regardless of the raw material."

5 Line 252-271, The results got from Figure 5 is not accurate. Significance analysis of difference should be adopted.  

3.3: "Regarding the yield of nanofibrillation (p = 2×10–7) or the transmittance (p = 10–4), the enzyme dosage imparted significant differences. However, increasing the dosage of endoglucanases from 80 to 240 mg/kg did not exert a notorious influence on SSA, and no significant influence was observed in the aspect ratio (p = 0.23)."

As stated now in the experimental part of the revised version, a difference is deemed significant if p<0.05, i.e., there is a low probability that the difference between the averages of the datasets is 0 (null hypothesis of paired t-tests).

Furthermore, the new Figure 5 is different and fortunately clearer than that from the original submission.

6 Line 281-284, I think Figure 5 should be Figure 6.

We thank the reviewer for pointing this out. The mistake has been corrected.

7 Line 290, I did not find Table 5

It referred to Table 4. This has been corrected in the revised version.

8 All data in the article should be analyzed based on the significance of diffrence to ensure the correct conclusion.

Paired t-tests have been performed in each case, reporting the p-value or its range.

Experimental, end of "2.4": "Datasets were compared in terms of paired t-tests to discern whether or not a difference is significant, opting for a significance level of α = 0.05 [34]. The null hypothesis is defined as follows: the average of the differences between two sets of values is 0. A significant difference is considered to be attained if the calculated p-value is less than 0.05."

3.1: "In any case, regarding the effects on fiber length, there was no significant difference between the different pretreatments (0.05 < p < 0.31), but all of them significantly shortened fibers."

3.2: "Likewise, the effect of the HPH steps was statistically independent of the raw material, finding no significant difference among jute, sisal, and hemp (0.26 < p < 0.83)."

3.4: "it could be said that MNFC from jute was found to be consistently more viscous than that from sisal (p = 8×10–5) and that from hemp (p = 0.015)"

9 Writing need to improve.

Some corrections have been made all along the manuscript: hyphenating when necessary ("large-scale" process, "worst-case" scenario); placing articles (at "an" industrial scale, such "a" phenomenon); approaches to conciseness ("whose scale-up is nothing short of challenging" -> "whose scale-up is challenging"), and aesthetic improvements ("that 'year' (...) along the following 'years'" -> "that year (...) over time").

Reviewer 3 Report

   In this manuscript, the authors investigated the effect of the starting materials (sisal, hemp, jute), pretreatment (mechanical, enzymatic), and the number of fibrillation cycles on the production of micro- and nanofibrillated cellulose (MNFC). As the results, it was revealed that all the factors affect the morphology, SSA, transparency, and viscosity of the products. In particular, it is interesting to conclude that the influence of differences in the starting materials cannot be ignored. On the other hand, it seems to be a bit unclear whether the difference was significant (is it reproducible?). In addition, further discussion on the origin of the differences due to the starting materials would be desirable. I think this manuscript may be suitable for Nanomaterials as one of the special issue "preparation, characterization and industrial application of Nanocellulose”, but it would be desirable to add the above discussion if possible.

   The followings are minor suggestions.

- In Figure 2c, the percentage value of cellulose is missing.

- In Table 4, it is helpful to add the explanation of k and n in the table.

Author Response

In this manuscript, the authors investigated the effect of the starting materials (sisal, hemp, jute), pretreatment (mechanical, enzymatic), and the number of fibrillation cycles on the production of micro- and nanofibrillated cellulose (MNFC). As the results, it was revealed that all the factors affect the morphology, SSA, transparency, and viscosity of the products. In particular, it is interesting to conclude that the influence of differences in the starting materials cannot be ignored. On the other hand, it seems to be a bit unclear whether the difference was significant (is it reproducible?). In addition, further discussion on the origin of the differences due to the starting materials would be desirable. I think this manuscript may be suitable for Nanomaterials as one of the special issue "preparation, characterization and industrial application of Nanocellulose”, but it would be desirable to add the above discussion if possible.

We want to thank the reviewer for considering our manuscript and for the helpful, constructive criticism. Admittedly, the original submission used terms like "significant" in a non-rigorous way, not following statistical calculations. Some Student t-tests have been incorporated. Regarding the discussion of how the properties measured can be related to the nature of the raw materials, the revised version has also been enriched in this direction — kindly find additions under "3.2" and "3.3".

The followings are minor suggestions.

In Figure 2c, the percentage value of cellulose is missing.

This mistake has been fixed ("86.5%").

In Table 4, it is helpful to add the explanation of k and n in the table.

Added: "the consistency factor (k) and the flow behavior index (n)"

Round 2

Reviewer 2 Report

The author has corrected and supplemented the manuscript.